# Continuous-Time Discrete Markov Bridge

## Abstract

Discrete diffusion has recently emerged as a promising paradigm in discrete data modeling. However, existing methods typically rely on a fixed-rate transition matrix during training, which not only limits the expressiveness of latent representations—a fundamental strength of variational methods—but also constrains the overall design space. To address these limitations, we propose **Discrete Markov Bridge**, a novel framework specifically designed for discrete representation learning. Our approach is built upon two key components: *Matrix*-learning and *Score*-learning. We conduct a rigorous theoretical analysis, establishing formal performance guarantees for *Matrix*-learning and proving the convergence of the overall framework. Furthermore, we analyze the space complexity of our method, addressing practical constraints identified in prior studies. Extensive empirical evaluations validate the effectiveness of the proposed **Discrete Markov Bridge**, which achieves an Evidence Lower Bound (ELBO) of **1.38** on the Text8 dataset, outperforming established baselines. Moreover, the proposed model demonstrates competitive performance on the CIFAR-10 dataset, achieving results comparable to those obtained by image-specific generation approaches.

## 1 Introduction

A fundamental question in generative modeling is estimating an underlying distribution, $\mu$, from observed data and subsequently generating new samples from this distribution. Among the various generative models proposed, diffusion models have exhibited remarkable performance in both continuous (Song et al., 2021; Ho et al., 2020b) and discrete domains (Campbell et al., 2022; Lou et al., 2024), demonstrating their versatility and effectiveness in diverse applications. These models effectively capture complex data distributions, enabling high-quality sample generation in various applications. However, despite their strong connection to variational models (Kingma & Welling, 2014; van den Oord et al., 2018), which are known for their impressive generative capabilities, diffusion models have yet to integrate the latent encoding ability inherent to variational approaches. Specifically, in the discrete domain, the noise rate transition matrices within discrete diffusion models are fixed and constrained, resulting in a limited design space and reduced expressive capacity. To the best of our knowledge, only the Absorb and Uniform Matrix (Campbell et al., 2022; Lou et al., 2024; Austin et al., 2021) have been considered in computations due to their simplicity in handling exponential term calculations.

In this study, we challenge the convention of using predefined static matrix in discrete modeling by introducing a novel approach, termed the **Discrete Markov Bridge (DMB)**, which aims to integrate the strengths of variational methods with discrete diffusion models, offering a more robust and efficient solution for complex discrete-state systems. This methodology seeks to enhance the modeling capabilities by leveraging the theoretical foundations of variational inference within the framework of discrete diffusion processes. Specifically, DMB is structured as a bidirectional two-stage learning algorithm. It comprises a forward variational process, i.e., *Matrix*-learning, that maps the data distribution to a learned distribution, followed by a backward decoding process, i.e., *Score*-learning, that reconstructs the data distribution from the learned representation.

In its matrix-learning process, DMB learns a rate transition matrix that maps the data distribution to an adapted noise distribution. A key feature of this matrix is its diagonalizability, which stands in stark contrast to the Absorb and Uniform matrices. This refinement enhances the model's adaptability and leads to improved performance. On the other hand, in the *Score*-learning process, a neural network is employed to model the concrete score (Lou et al., 2024; Meng et al., 2023). As for the sampling

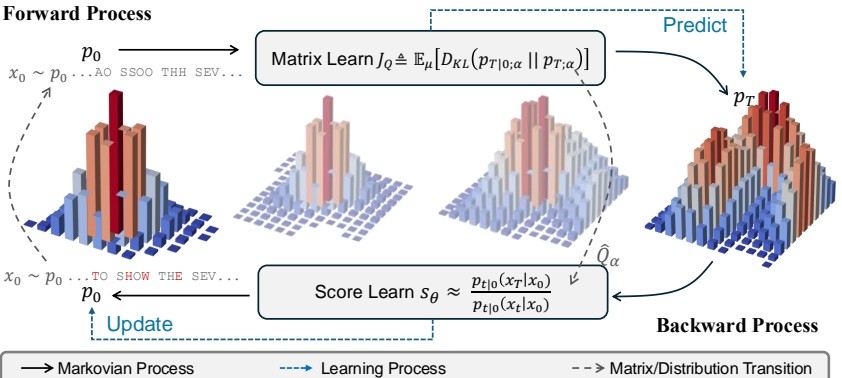

Figure 1: **Discrete Markov Bridge** (**DMB**) comprises two components: *Matrix*-learning and *Score*-learning. The *Matrix*-learning stage learns an adaptive transition-rate matrix to estimate the noise-adapted distribution. This predicted distribution supervises the *Score*-learning stage via its loss. In parallel, the *Score*-learning stage estimates the probability ratio required to construct the inverse transition-rate matrix, enabling reconstruction of the original data distribution.

procedure, the rate transition matrix derived from the *Matrix*-learning process and the neural network obtained from the *Score*-learning process are jointly employed to solve the backward differential equation.

Within this framework, a broad spectrum of tasks can be effectively addressed. For discrete data modalities such as text, the model supports non-autoregressive generation, following the approach outlined in (Gu & Tan, 2022). In this work, we demonstrate that our proposed method surpasses the performance of the previously established SEDD model (Lou et al., 2024). For image data, the model can be integrated with a VQ-VAE architecture (van den Oord et al., 2018), yielding performance on par with that of DDPM when evaluated on the CIFAR-10 dataset.

We summarize our contributions as follows:

- **Novel Framework for Discrete Data (Section 3):** We introduce the Discrete Markov Bridge, a new variational framework for modeling complex discrete data.
- **Theoretical Guarantee (Section 4):** We offer a theoretical guarantee in terms of the validity and accessibility of *Matrix*-learning as well as the convergence of the whole framework.
- **Addressing Practical Issues (Section 5):** Building on the theoretical insights established earlier, we propose a computationally efficient matrix to tackle the practical challenges discussed in Section 5. We then evaluate the model's performance through experiments, demonstrating that it outperforms baseline methods in text modeling and provides comparable image modeling results.

## 2 PRELIMINARIES AND RELATED WORKS

### 2.1 CONTINUOUS-TIME DISCRETE MARKOV CHAIN

Let $\mathbb{X} = \{1, 2, \ldots, n\}$ denote a finite state space, where $n \in \mathbb{R}$. A continuous time discrete Markov chain (CTDMC) defined on $\mathbb{X}$ is represented as $\{X(t) \mid t \in \mathbb{R}, X(t) \in \mathbb{X}\}$. For convenience, we use the notation $X_t \triangleq X(t)$. The probability of transitioning from state $x \in \mathbb{X}$ at time $t$ to state $y \in \mathbb{X}$ at time $t + s$ is denoted as $p_{t+s|t}(y|x) \triangleq P(X_{t+s} = y \mid X_t = x)$. Similarly, the probability that $X_t$ takes state $x$ at time $t$ is expressed as $p_t(x) \triangleq P(X_t = x)$. The probability distribution over the state space at time $t$ is then given by the vector $p_t \triangleq (p_t(1), p_t(2), \ldots, p_t(n))$. The core component to describe a continuous time discrete Markov chain is the rate transition matrix. We defined the rate transition probability as follows:

$$q_t(x, y) \triangleq \frac{dp_{t+s|t}(y|x)}{ds} = \lim_{\Delta s \to 0} \frac{p_{t+s|t}(y|x) - p_{t|t}(y|x)}{\Delta s} = \lim_{\Delta s \to 0} \frac{p_{t+s|t}(y|x) - \delta_x(y)}{\Delta s},$$

where $\delta_x(y)$ is the Dirac delta function. The Forward Kolmogorov Equation can be written as $\frac{dp_t}{dt} = p_t Q^{(t)}$. $Q_{x,y}^{(t)} \triangleq q_t(x, y)$, for all $x, y \in \mathbb{X}$, denotes the rate transition matrix at time $t$. The subscripts $x$ and $y$ indicate the row and column indices, respectively. Each rate transition matrix satisfies the conditions: the sum of each row must be zero, and all off-diagonal entries must be non-negative. Formally, this is expressed as $\sum_y Q_{x,y} = 0$ for all $x$ and $Q_{x,y} \geq 0$ for all $y \neq x$.

## 2.2 RELATED WORKS

**Prior Learning** Leveraging a prior is a longstanding paradigm in machine learning. In the field of natural language processing, for example, training typically begins with pretrained language models (Liu et al., 2019; Devlin et al., 2019; Touvron et al., 2023; Radford et al., 2019; Lan et al., 2020; Hu et al., 2021; Vaswani et al., 2023). Likewise, pretrained models are highly valued in computer vision (He et al., 2015). In our approach, the concept of a prior is equally fundamental: the forward process adaptively refines this prior based on the evolving training dynamics of the backward process.

**Discrete Diffusion Models** Diffusion models (Ho et al., 2020b; Song et al., 2021; 2022; Sohl-Dickstein et al., 2015) add noise to data and use a denoiser for reconstruction, achieving success in image tasks and gaining traction in discrete domains like natural language (Li et al., 2022; Lou et al., 2024; Campbell et al., 2022; Gulrajani & Hashimoto, 2023; Sun et al., 2023; Dieleman et al., 2022; Nie et al., 2025). Some methods map discrete data to continuous space (Li et al., 2022; Gulrajani & Hashimoto, 2023), introducing rounding errors, while others operate directly in discrete space but impose rigid, non-learnable noise structures (Campbell et al., 2022; Lou et al., 2024). In the continuous domain, trainable Gaussian parameters improve flexibility (Kingma et al., 2023), but no such method exists for discrete diffusion, where Gaussian distributions also remain restrictive. Moreover, masked discrete diffusion models struggle to learn temporal dependencies (Zheng et al., 2024).

**Flow Models** Flow-based models (Rezende & Mohamed, 2016; Kingma & Dhariwal, 2018; Liu et al., 2022; Satorras et al., 2022; Albergo et al., 2023; Trockman & Kolter, 2021) constitute a prominent class of machine learning models characterized by their ability to perform reversible transformations on data representations. In contrast to conventional flow models, which rely on transformation paths predefined by human designers (Albergo et al., 2023; Liu et al., 2022), our approach autonomously learns these paths, enhancing adaptability and expressiveness in data modeling.

## 3 DISCRETE MARKOV BRIDGE

The target distribution, denoted as $\mu \in \mathbb{R}^n$, is a probability vector, meaning that its elements are non-negative and collectively sum to one. As shown in Figure 1, our objective is to estimate the distribution at one endpoint of the Markov chain, denoted as $p_0$, such that $p_0 \approx \mu$. The other endpoint, denoted as $p_T$, serves as the distribution for the latent variables or prior. To achieve the specified objectives, the proposed **DMB** framework is structured into two distinct components: *Matrix Learning* and *Score Learning*.

The *Matrix*-learning serves as a forward bridge, facilitating the transition from $\mu$ to the latent distribution. Conversely, the *Score*-learning function represents a reverse pathway from the latent distribution back to $\mu$, leveraging the groundwork established by the *Matrix*-learning process. This dual-function framework ensures a comprehensive bidirectional understanding of the data structure in a variational-like manner.

The structure of the **DMB** is demonstrated in Algorithm 2. This pseudocode consists of two nested while loops that operate within the overarching while loop governing the training epochs. Each of these nested loops corresponds to a distinct learning stage within the framework. We list the following theorem to ensure the reversibility of the forward and backward Markovian processes.

**Theorem 3.1** (Reversibility (Campbell et al., 2022; Lou et al., 2024))**.** *Given the Forward Kolmogorov Equation of a CTDMC:*

$$\frac{dp_t}{dt} = p_t Q^{(t)} \tag{1}$$

*There exists a reverse CTDMC with Forward Kolmogorov Equation:*

$$\frac{dp_{T-t}}{dt} = p_{T-t}\hat{Q}^{(T-t)} \text{ ,where } \hat{Q}_{x,y}^{(t)} = \frac{p_t(y)}{p_t(x)}Q_{y,x}^{(t)} \tag{2}$$

This theorem elucidates the reverse form of a CTDMC, proposing that knowledge of the probability ratio enables the derivation of a reversal of the original Markov chain that is almost everywhere equivalent. The derivation of the reverse form underscores the theoretical framework that mirrors the dynamics of the forward stochastic process.

We fomulate the learning process by employing the continuous-time Evidence Lower Bound (ELBO) as an alternative optimization objective to Maximum Likelihood Estimation (MLE). In the **DMB** framework, both *Matrix*-learning and *Score*-learning collaboratively optimize the full bound through their respective subprocesses.

### 3.1 *Matrix*-LEARNING

In the *Matrix*-learning process, our primary objective is to estimate the rate transition matrix $Q_\alpha$, where $\alpha$ denotes the set of model parameters. For simplicity, we assume that the forward rate transition matrix at time $t$, denoted $Q_\alpha^{(t)}$, is given by $\sigma(t)Q_\alpha$. Furthermore, we employ the following $Q_\alpha$:

$$Q_\alpha = A \begin{bmatrix} -\sum_{i=1}^{n-1} a_i & a_1 & \dots & a_{n-2} & a_{n-1} \\ 0 & -\sum_{i=2}^{n-1} a_i & \dots & a_{n-2} & a_{n-1} \\ \dots & \dots & \dots & \dots & \dots \\ 0 & 0 & \dots & -a_{n-1} & a_{n-1} \\ 0 & 0 & \dots & 0 & 0 \end{bmatrix} A^{-1} := AHA^{-1} \tag{3}$$

, where $\{a_1, a_2, \dots, a_{n-1}\} = \alpha$ are parameters for learning, $A, A^{-1}$ are fixed predefined permutation matrices and $H$ is denoted as the upper-triangle matrix in the equation. The derivation and underlying rationale for utilizing this matrix are detailed in Section 4 and further explored in Section 5.1. Another essential component of this process is $\mu$, which is approximated using the currently predicted $p_0$ obtained through *Score*-learning as a prior (see Section 3.2). By integrating Equation (1) from time 0 to time $t$, the following equation can be derived:

$$p_t = p_0 \exp\{\int_0^t \sigma(s)dsQ_\alpha\} \tag{4}$$

Note that the exponential in the formula is a matrix exponential. The training procedure aims to minimize a component of the variational bound (see Equation (9)), leading to the following objective function $J_Q$:

$$J_Q \triangleq \mathbb{E}_\mu[D_{KL}(p_{T|0;\alpha}||p_{T;\alpha})], \tag{5}$$

where the conditional probability distribution $p_{T|0;\alpha}$ is given by the rows of $\exp\{\int_0^T \sigma(s)dsQ_\alpha\}$:

$$p_{T|0;\alpha}(x_T|x_0) = \exp\{\int_0^T \sigma(s)dsQ_\alpha\}_{x_0,x_T} \tag{6}$$

The final distribution $p_{T;\alpha}$ is obtained by multiplying the initial distribution $p_0$ with the conditional distribution, as presented in Equation (4), evaluated at time $t = T$.

---

**Algorithm 2** Training Algorithm of the **DMB**

---

**Input:** Target discrete data $X \sim \mu$

1: Initialize $p_0, p_T \leftarrow random\_init()$
2: **while** *not converge* **do**
3:    Sample a batch of discrete instance $X_0 \sim \mu$. */* Data for the two learning processes. */*
     */* Matrix Learning */*
4:    $step \leftarrow 0$
5:    **while** $step \leq max\_step$ & $\mathcal{L}_Q \geq \epsilon_Q$ **do**
6:      Update $Q_\alpha, \mathcal{J}_Q$ according to Eqn. (5) and predict $p_T$ using Eqn. (4) at $t = T$.
7:      $step \leftarrow step + 1$
8:    **end while**
     */* Score Learning */*
9:    $step \leftarrow 0$
10:   **while** $step \leq max\_step$ & $\mathcal{J}_{score} \geq \epsilon_{score}$ **do**
11:      Update $s_\theta, \mathcal{J}_{score}$ *w.r.t.* current $Q_\alpha$ using Eqn. (8).
12:      $step \leftarrow step + 1$
13:   **end while**
14:   Predict updated $p_0$ that estimates $\mu$ using Eqn. (10). */* Used for Matrix Learning */*
15:   **if** $\mathcal{J}_Q + \mathcal{J}_{score} < \epsilon$ **then**
16:      $converge \leftarrow$ TRUE
17:   **end if**
18: **end while**

---

### 3.2 *Score*-LEARNING

*Score*-learning constitutes a reverse process of *Matrix*-learning. It is noted that in Theorem 3.1, the reverse rate transition matrix adheres to the following relationship:

$$\hat{Q}_{x,y}^{(t)} = \frac{p_t(y)}{p_t(x)} Q_{x,y} \sigma(t) \tag{7}$$

Consequently, while *Matrix*-learning handles the forward rate transition matrix $Q$, *Score*-learning focuses on managing the remaining part, i.e $\frac{p_t(y)}{p_t(x)}$ (Lou et al., 2024). A learnable model $s_\theta(x_t, t)_y$ is designed to model the ratio, and the main part of the continuous time Evidence Lower Bound (ELBO) (Campbell et al., 2022; Lou et al., 2024; Kingma & Welling, 2014) is leveraged as the training objective, denoted as $J_{score}$:

$$\int_0^T \mathbb{E}_{x_0 \sim \mu, x_t \sim p_{t|0}} \Big[ \sum_{y \neq x_t} Q_{y,x_t}^{(t)} \Big( s_\theta(x_t, t)_y - \frac{p_{t|0}(y|x_0)}{p_{t|0}(x_t|x_0)} + \frac{p_{t|0}(y|x_0)}{p_{t|0}(x_t|x_0)} \Big( \log(\frac{p_{t|0}(y|x_0)}{p_{t|0}(x_t|x_0)}) - \log s_\theta(x_t, t)_y \Big) \Big) \Big] dt \tag{8}$$

To provide a comprehensive understanding, we present the complete ELBO as follows, demonstrating how *Matrix*-learning and *Score*-learning collaboratively contribute to minimizing the ELBO bound.

$$\mathbb{E}_{x_0 \sim \mu}[-\log p_{0;\theta}(x_0)] \leq J_{score} + J_Q. \tag{9}$$

**Estimating** $\mu$   The estimation of $\mu$ is expressed as Equation (10). The equation below is derived under the Euler method and can be generalized to other ODE-solving methods. Suppose the inference time process is partitioned as: $[0, t_1], [t_1, t_2], \ldots, [t_n, T]$. By Baye's rules:

$$\mu(x_0) \approx p_0(x_0) = \mathbb{E}_{X_T, X_n, \ldots X_1}[p_{0|1}(x_0|x_1)]. \tag{10}$$

Under the guidance of Equation (10), the sampling process begins with drawing $x_T$, followed by obtaining $x_n$ through the conditional distribution $p_{t_n|T}(x_n|X_T = x_T)$. This procedure continues iteratively, generating $x_{n-1}$, and proceeding sequentially until the complete sequence $\{X_T, X_n, \ldots, X_1\}$ is sampled. Subsequently, the conditional probability $p_{0|1}(x_0|x_1)$ is determined. By repeating this process multiple times and averaging the sampled probabilities, an estimation can be obtained by approximating the expectation with the empirical mean.

## 3.3 SAMPLING

The sampling process is done under the cooperation of *Matrix*-learning and *Score*-learning in a similar way as estimating $\mu$. The reverse rate transition matrix is calculated as Equation (7), and an ode-solving method such as the Euler method can be further applied to solve Equation (2). Noticed that, as shown in line 15 of Algorithm 2, the sampling process is performed every time after the *Score*-learning process to gain the estimation of $\mu$ and samples for evaluation.

## 4 THEORETICAL FOUNDATIONS

### 4.1 VALIDITY AND ACCESSIBILITY OF *Matrix*-LEARNING

In this subsection, we establish the validity and accessibility of the Matrix-learning process. Specifically, **validity** concerns the ultimate state of the forward process and whether it remains confined within a well-defined domain, i.e., whether a probability distribution transforms into another valid probability distribution. **Accessibility**, on the other hand, pertains to the ability of the process to transition between any two arbitrary discrete distributions.

**Validity**    Proposition 4.1, presented below, establishes that any transformation originating from a probability distribution must result in another probability distribution. This theorem guarantees that, despite the presence of errors in the learning process, the outcome remains a valid probability distribution. For a detailed proof, refer to Appendix A.

**Proposition 4.1** (Conservation of the Sum). *For two arbitrary vectors* $\phi, \mu \in \mathbb{R}^n$, *rate transition matrix* $Q \in \mathcal{R}^{n \times n}$, *if* $\phi = \mu \exp\{Q\}$, *then*

$$\sum_{i=1}^{n} \phi[i] = \sum_{i=1}^{n} \mu[i]$$

**Accessibility**    Theorem 4.2 ensures that any two probability distributions are accessible in the forward process. Consequently, this implies that the optimality of *Matrix*-learning can be achieved, provided the presence of a strong optimizer.

**Theorem 4.2** (Accessibility). *For two arbitrary discrete distributions* $p, q \in \mathbb{R}^n$, *there exists a rate transition matrix* $Q \in \mathbb{R}^{n \times n}$ *such that:*

$$p = qe^Q \tag{11}$$

The central idea of the proof is to construct a specialized matrix that possesses strong representational capacity while remaining computationally manageable within the framework of matrix exponentiation. The designed matrix, which is depicted in Lemma 4.3, is an upper triangle matrix with the vanished sum of rows. A remarkable characteristic of this matrix is its elegant eigendecomposition form, which presents a well-structured and analytically convenient representation. Its eigenmatrix is an all-one upper triangular matrix, as shown in Lemma 4.3.

**Lemma 4.3.** *Let matrix* $Q = H \in \mathbb{R}^{n \times n}$, *where* $H$ *is defined in Equation* (3), *then* $Q$ *can be diagonalized as* $Q = U\Lambda U^{-1}$, *where and the diagonal matrix is* $\Lambda = diag(\{-\sum_{i=1}^{n-1} a_i, -\sum_{i=2}^{n-1} a_i, \ldots, -a_{n-1}, 0\})$, *the orthoganl matrix is* $U = \begin{bmatrix} 1 & 1 & \ldots & 1 \\ 0 & 1 & \ldots & 1 \\ \ldots & \ldots & \ldots & \ldots \\ 0 & 0 & \ldots & 1 \end{bmatrix}$.

There are two key observations regarding the $Q$ matrix. First, it contains only $n-1$ parameters, which constitute the minimal set necessary to solve Equation (11). This sufficiency implies that the solution derived for the $Q$ matrix is unique. Second, the matrix retains nonzero elements exclusively in its upper triangular portion, implying that each element can transition only to those with a larger index. This observation raises an additional consideration: for effective state transitions, the matrix must allocate sufficient "mass" or probability. Consequently, a matrix is required to appropriately adjust the indices of elements within the finite set $\mathbb{X}$, as shown in Lemma 4.4. Lemma 4.4 establishes that, after a permutation, the cumulative probability at each element of the initial distribution in the transition process is greater than or equal to that of the target distribution. This guarantees that

elements with surplus probability can redistribute their excess, while those with a deficiency can receive the necessary adjustments, ensuring a balanced transformation.

**Lemma 4.4.** *For arbitrary distribution $p, q \in \mathbb{R}^n$, there exists an permutation matrix $A$ such that:*

$$\frac{p'_1}{q'_1} \leq \frac{p'_1 + p'_2}{q'_1 + q'_2} \leq \dots \leq \frac{\sum_{i=1}^{k} p'_i}{\sum_{i=1}^{k} q'_i} \leq \dots \leq \frac{\sum_{i=1}^{n} p'_i}{\sum_{i=1}^{n} q'_i} = 1 \tag{12}$$

*where $p' = pA$, $q' = qA$, $p'_i$ is the $i$-th entry of $p'$, and $q'_i$ is the $i$-th entry of $q'$.*

This lemma guarantees the existence of a permutation that, when applied, results in a monotonically increasing sequence of probabilities.

**Lemma 4.5.** *Let $Q \in \mathbb{R}^{n \times n}$ be a rate transition matrix, $A \in \mathbb{R}^{n \times n}$ be a permutation matrix, then $AQA^{-1}$ is a rate transition matrix.*

By integrating the lemmas above, we aim to establish the proof of Theorem 4.2. A comprehensive derivation of these lemmas and the theorem is provided in Appendix B.

## 4.2 CONVERGENCE

As discussed earlier, the **DMB** framework operates as a two-step learning algorithm, necessitating a thorough examination of its convergence properties. In this section, we present a theorem that establishes the convergence of the entire algorithm. The convergence problem is nontrivial, as the *Score*-learning process does not merely constitute a direct inversion of the *Matrix*-learning process. The discrepancy arises because the score model $s_\theta$ is trained under the supervision of the distribution $\mu$, rather than $p_0^{(k)}$, where $k$ denotes the epoch number. To be specific, we have

**Proposition 4.6** (Supervision of *Score*-learning). *Suppose $Q_t$'s elements are non-zeros, the training objective is depicted as in Equation (8), then the optimal score model $s_{\theta^*}(x_t, t)_y$ satisfies:*

$$s_{\theta^*}(x_t, t)_y = \mathbb{E}_{x_0 \sim \mu_{0|t}(\cdot|x_t)}\left[\frac{p_{t|0}(y|x_0)}{p_{t|0}(x_t|x_0)}\right] = \frac{\sum_{x_0} \mu(x_0) p_{t|0}(y|x_0)}{\sum_{x_0} \mu(x_0) p_{t|0}(x_t|x_0)}$$

The proposition presented above illustrates the influence of $\mu$ on the training process and underscores the challenge of convergence arising from the absence of $p_0^{(k)}$. A detailed proof of this proposition can be found in Appendix C.

Under the assumption that each process achieves optimality, the following theorem establishes the convergence of **DMB** from the perspective of KL divergence, thereby demonstrating the validity of the overall **DMB** framework. Moreover, given our primary focus on the algorithmic aspects, this assumption is justified, consistent with prior work that introduces new frameworks, such as Goodfellow et al. (2014). Notably, although the training objective of the *Score*-learning process is the continuous ELBO bound, the theorem presented below can be generalized to encompass a broader class of objectives. This generalization suggests the potential for designing improved training objectives within our framework.

**Theorem 4.7** (Convergence of the algorithm). *If we assume optimality is achieved in every epoch of the Matrix-learning process and the Score-learning process, and we denote the $k$-th epoch estimation of $\mu$ as $p_0^{(k)}$, then $\lim_{k \to \infty} D_{KL}(\mu || p_0^{(k)})$ converges.*

Please refer to Appendix D for the proof.

## 5 PRACTICAL ISSUES AND EXPERIMENTS

In this section, we discuss the practical issues of **DMB** by assuming our data coming from a high dimensional space, i.e. $\mu \in \mathbb{R}^{d \times n}$, where $n$ is the size of the finite set and $d$ is the number of dimensions. For instance, for textual data, $n$ is the size of the vocabulary and $d$ is the sequence length.

## 5.1 Addressing Practical Issues

Both the **DMB** model and discrete diffusion models (Lou et al., 2024; Campbell et al., 2022; Sun et al., 2021) face significant challenges related to the $Q$ matrix. In particular, during the *Score*-learning process, the computational efficiency of matrix exponential operations becomes a critical constraint. Furthermore, the *Matrix*-learning process often requires storing the entire $Q$ matrix, posing substantial concerns regarding space efficiency. These limitations have been the primary reasons restricting previous studies to utilizing only the Uniform and Absorb matrices.

As Jean le Rond d'Alembert once remarked, *Algebra is generous; she often gives more than is asked of her.* In the context of proving Theorem 4.2, we identify a distinct class of matrices, as mentioned in Section 3.1 and further rigorously discussed in Lemma 4.3. This structured approach not only underscores the theoretical underpinnings but also highlights the practical implications of matrix manipulation in these models.

**Efficient Computation of the permutation matrix.** Before proceeding with the analysis of the $Q_\alpha$ matrix, we first outline the computation of the predefined permutation matrix $A$. As illustrated in the assumptions, the evolution of each dimension occurs independently. Consequently, for each dimension, the permutation matrix is computed separately. In accordance with Lemma 4.4, we assume the denominator to be constant. Therefore, the permutation matrix for the $i$-th dimension satisfies the following inequality:

$$\mu(X_0^{(i)} = j) \leq \mu(X_0^{(i)} \leq j+1), \forall j \in 1, 2, \ldots, n-1$$

The marginal distribution $\mu(X_0^{(i)})$ can be efficiently estimated in the form of a histogram by extracting a subbatch from the dataset. Subsequently, the permutation matrix is computed using a fast sorting algorithm with a time complexity of $O(n \log n)$.

**Efficient Computation of Matrix Exponential.** Matrix exponential is difficult to calculate as it's defined through Taylor expansion, however, a property exists:

**Proposition 5.1.** *For a matrix $Q \in \mathbb{R}^{n \times n}$ and a non-degenerate matrix $D \in \mathbb{R}^{n \times n}$, we have* $\exp\{DQD^{-1}\} = D \exp\{Q\}D^{-1}$

Please refer to Appendix E for proof. By Proposition 5.1,

$$\exp\{Q_\alpha\} = \exp\{(AU)\Lambda_\alpha(AU)^{-1}\} = (AU)\exp\{\Lambda_\alpha\}(AU)^{-1} \tag{13}$$

, where $U$ is the all-one upper triangle matrix, $\Lambda_\alpha$ is a diagonal matrix parameterized by $\alpha$. Therefore, the computation of the matrix exponential is reduced to evaluating the exponential of a diagonal matrix, which is significantly more efficient.

**Space Efficiency.** For the permutation matrices $A, A^{-1} \in \mathbb{R}^{d \times n \times n}$, a total of $d \times 2n$ parameters are required. Apart from $A, A^{-1}$, the upper triangle matrix can be decomposed into a non-parameterized all-one upper triangle matrix, a parameterized diagonal matrix, and a constant matrix. Consequently, the storage requirement is of the order $O(nd)$ parameters.

## 5.2 ELBO Bound Calculation

As shown in Equation (9), the computation of the full bound necessitates the evaluation of both the $J_{score}$ and the expected Kullback–Leibler (KL) divergence between the evolved distribution and the target distribution, expressed as $\mathbb{E}_\mu D_{KL}(P_{T|0}||P_T)$. Under mild assumptions following previous works (Campbell et al., 2022; Lou et al., 2024), we can derive a closed-form expression for computing the KL term:

**Proposition 5.2.** *The KL term can be calculated as:*

$$D_{KL}\big(p_{T|0;\alpha}(x_T|x_0)||p_T(x_T)\big) = \sum_{i=1}^{d} D_{KL}\big(p_{T|0;\alpha}(x_T^{(i)}|x_0^{(i)})||p_T(x_T^{(i)})\big) \tag{14}$$

## 5.3 EXPERIMENT

Table 1: The results were tested 1000 times on the Text8 dataset. We adopt the baseline results reported in (Lou et al., 2024) for comparison. AR: Autoregressive generation. NAR: Non-autoregressive generation.

| Model | BPC ($\downarrow$) |
|---|---|
| *AR* | |
| IAF/SCF (Ziegler & Rush, 2019) | 1.88 |
| AR Argmax Flow (Hoogeboom et al., 2021) | 1.39 |
| Discrete Flow (Tran et al., 2019) | 1.23 |
| *NAR* | |
| SEDD Absorb (Lou et al., 2024) | $\leq 1.39$ |
| D3PM Absorb (Austin et al., 2021) | $\leq 1.45$ |
| Mult. Diffusion (Hoogeboom et al., 2021) | $\leq 1.72$ |
| MAC (Shih et al., 2022) | $\leq 1.40$ |
| BFN (Graves et al., 2024) | $\leq 1.41$ |
| *NAR: w/o [mask] token* | |
| D3PM Uniform (Austin et al., 2021) | $\leq 1.61$ |
| SEDD Uniform (Lou et al., 2024) | $\leq 1.47$ |
| **DMB** (Ours) | $\leq \mathbf{1.38}$ |

Table 2: CIFAR-10 Results. We report inception score (IS), and Fréchet Inception Distance (FID) score. Results are adopted from Ho et al. (2020a).

| Model | IS ($\uparrow$) | FID ($\downarrow$) |
|---|---|---|
| **Conditional** | | |
| EBM (Du & Mordatch, 2019) | 8.30 | 37.9 |
| JEM (Grathwohl et al., 2020) | 8.76 | 38.4 |
| BigGAN (Brock et al., 2019) | 9.22 | 14.73 |
| StyleGAN2 + ADA (v1) (Karras et al., 2020) | **10.06** | **2.67** |
| **Unconditional** | | |
| Gated PixelCNN (van den Oord et al., 2016) | 4.60 | 65.93 |
| PixelIQN (Ostrovski et al., 2018) | 5.29 | 49.46 |
| EBM (Du & Mordatch, 2019) | 6.78 | 38.2 |
| NCSN (Song & Ermon, 2019) | 8.87±0.12 | 25.32 |
| SNGAN (Miyato et al., 2018) | 8.22±0.05 | 21.7 |
| SNGAN-DDLS (Che et al., 2020) | 9.09±0.10 | 15.42 |
| StyleGAN2 + ADA (v1) (Karras et al., 2020) | **9.74** ± 0.05 | 3.26 |
| DDPM (fixed isotropic) (Ho et al., 2020a) | 7.67±0.13 | 13.51 |
| DDPM (simple) (Ho et al., 2020a) | 9.46±0.11 | **3.17** |
| **Discrete Diffusion** | | |
| D3PM (absorbing) (Austin et al., 2021) | 6.78 | - |
| D3PM (uniform) (Austin et al., 2021) | 5.99 | - |
| D3PM (Gauss + Logistic) (Austin et al., 2021) | 8.56 | - |
| **Ours** | **8.64** | **11.63** |

**Best Performance on Text8** The proposed framework was evaluated on the Text8 dataset, with experimental results summarized in Table 1. Performance was measured using the Evidence Lower Bound (ELBO), calculated as detailed in Section 5.2, and results were averaged over 1,000 independent trials to ensure statistical reliability. Our model, **DMB**, achieves a Bits-Per-Character (BPC) of 1.38, outperforming representative discrete diffusion baselines such as SEDD (Lou et al., 2024). Notably, this performance is achieved without introducing a mask token or otherwise modifying the vocabulary. When compared to other methods that also operate without a mask token, such as SEDD Uniform and D3PM Uniform, our approach demonstrates a significant performance gain, improving upon previous SOTA by approximately **0.1 BPC**.

**Competitive Performance on CIFAR-10** We evaluated the performance of **DMB** on the CIFAR-10 dataset within a VQ-VAE framework (van den Oord et al., 2018). The quantitative results, presented in Table 2, show that our model achieves an Inception Score (IS) of 8.64 and a Fréchet Inception Distance (FID) of 11.63. This performance not only surpasses that of another discrete diffusion model, D3PM, by 0.08 in IS but also outperforms several models explicitly designed for image generation—including DDPM (fixed isotropic) and SNGAN (Miyato et al., 2018)—across both IS and FID metrics. The primary advantage of our method over D3PM stems from its core mechanism: whereas D3PM relies on fixed rate transition matrices (e.g., Absorb, Uniform, Gauss), **DMB** adaptively learns the optimal matrix, leading to better outcomes. Collectively, these results underscore the effectiveness and generalization capability of our model, demonstrating its strong performance even beyond its primary design scope.

## 6 CONCLUSION

In this study, we propose a novel paradigm, the **Discrete Markov Bridge** (**DMB**), which combines the strengths of variational methods with the capabilities of discrete diffusion models. We provide theoretical guarantees to substantiate the validity and accessibility of the proposed *Matrix*-learning process and further prove the convergence of the DMB algorithm. In addition to our theoretical contributions, we conduct extensive empirical evaluations on the Text8 and CIFAR-10 datasets. The experimental results indicate that **DMB** not only surpasses existing baselines such as SEDD Lou et al. (2024) in text modeling tasks, but also achieves competitive performance in image modeling on CIFAR-10, thereby demonstrating its potential as a unified framework for discrete representation learning.

## ETHICS STATEMENT

This research is strictly methodological, focusing on foundational algorithmic improvements for discrete diffusion. Our work does not involve human participants, sensitive data, or real-world deployment. As we introduce no new datasets, our contributions do not raise concerns regarding data privacy, bias, or potential misuse. While we acknowledge the broader societal implications of language models in downstream applications, our study is confined to a pre-deployment, academic context. The enhancements detailed here are not designed for and do not facilitate manipulation, deception, or other unethical activities. We therefore conclude that our research poses no direct ethical or societal risks and is aligned with the principles of responsible AI development.

## REPRODUCIBILITY STATEMENT

To ensure our work is fully reproducible, we provide a comprehensive overview of our methods and experiments (See Sections 3 and 5). The technical formulation of our approach is detailed in Section 3, while the experimental details are listed in Appendix G. Our LLM usage statement is located in Appendix I, and the complete source code is included with the submission to facilitate replication. We also attach the code in the supplementary materials.

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

# Appendix

## Contents

## A    PROOF OF CONSERVATION OF THE SUM

**Proposition A.1** (Conservation of the Sum). *For two arbitrary vectors $\phi, \mu \in \mathbb{R}^d$, rate transition matrix $Q \in \mathcal{R}^{d \times d}$, if $\phi = \mu \exp^Q$, then*

$$\sum_{i=1}^{d} \phi[i] = \sum_{i=1}^{d} \mu[i]$$

*Proof.* As $\phi = \mu \exp^Q$,

$$\phi(i) = \sum_j \mu(j)(\exp\{Q\})_{j,i}$$

Therefore,

$$\sum_i \phi(i) = \sum_i \sum_j \mu(j)(\exp\{Q\})_{j,i}$$

As we have

$$\sum_j (\exp\{Q\})_{i,j} = 1$$

Thus,

$$\sum_i \phi(i) = \sum_j \mu(j) \sum_i (\exp\{Q\})_{j,i} = \sum_j \mu(j)$$

$\blacksquare$

## B    PROOF OF ACCESSIBILITY

### B.1    PROOF OF LEMMAS

**Lemma B.1.** *Let matrix $Q \in \mathbb{R}^{d \times d}$ and hold the following form:*

$$Q = \begin{bmatrix} -\sum_{i=1}^{n-1} a_i & a_1 & a_2 & \ldots & a_{n-2} & a_{n-1} \\ 0 & -\sum_{i=2}^{n-1} a_i & a_2 & \ldots & a_{n-2} & a_{n-1} \\ 0 & 0 & -\sum_{i=3}^{n-1} a_i & \ldots & a_{n-2} & a_{n-1} \\ \ldots & \ldots & \ldots & \ldots & \ldots & \ldots \\ 0 & 0 & 0 & \ldots & -a_{n-1} & a_{n-1} \\ 0 & 0 & 0 & 0 & 0 & 0 \end{bmatrix}$$

*then $Q$ can be diagonalized in the following form:*

$$Q = U \Lambda U^{-1}$$

*,where $U = \begin{bmatrix} 1 & 1 & 1 & \ldots & 1 \\ 0 & 1 & 1 & \ldots & 1 \\ 0 & 1 & 1 & \ldots & 1 \\ \ldots & \ldots & \ldots & \ldots & \ldots \\ 0 & 0 & 0 & \ldots & 1 \end{bmatrix}, \Lambda = diag(\{-\sum_{i=1}^{n-1} a_i, -\sum_{i=2}^{n-1} a_i, \ldots, -a_{n-1}, 0\})$*

*Proof.* $Q = \begin{bmatrix} 1 & 1 & 1 & \ldots & 1 \\ 0 & 1 & 1 & \ldots & 1 \\ \ldots & \ldots & \ldots & \ldots & \ldots \\ 0 & 0 & \ldots & 1 & 1 \\ 0 & 0 & 0 & 0 & 1 \end{bmatrix} diag(\{-\sum_{i=1}^{n-1} a_i, -\sum_{i=2}^{n-1} a_i, \ldots, -a_{n-1}, 0\}) \begin{bmatrix} 1 & -1 & 0 & \ldots & 0 \\ 0 & 1 & -1 & \ldots & 0 \\ \ldots & \ldots & \ldots & \ldots & \ldots \\ 0 & 0 & \ldots & 1 & -1 \\ 0 & 0 & 0 & 0 & 1 \end{bmatrix}$

$\blacksquare$

**Lemma B.2.** *For arbitrary distribution $p, q \in \mathbb{R}^{1 \times d}$, there exists an permutation matrix $A$ such that:*

$$\frac{p'_1}{q'_1} \leq \frac{p'_1 + p'_2}{q'_1 + q'_2} \leq \cdots \leq \frac{\sum_{i=1}^{k} p'_i}{\sum_{i=1}^{k} q'_i} \leq \cdots \leq \frac{\sum_{i=1}^{n} p'_i}{\sum_{i=1}^{n} q'_i} = 1 \tag{15}$$

*where $p' = pA$, $q' = qA$, $p'_i$ is the $i$-th entry of $p'$*

*Proof.* It's obvious that there exists a permutation matrix $A$ which can sort $\frac{p_i}{q_i}$ ascendly, i.e.:

$$\frac{p'_i}{q'_i} \leq \frac{p'_{i+1}}{q'_{i+1}}$$

, where $p' := pA, q' := qA$, and the corner mark $i$ refer to the $i$-th entry.

Also, we can demonstrate that:

$$\frac{a_1}{b_1} \leq \frac{a_2}{b_2} \Rightarrow \frac{a_1}{b_1} \leq \frac{a_1 + a_2}{b_1 + b_2} \leq \frac{a_2}{b_2} \tag{$\triangle$}$$

The inequality we need to prove is:

$$\frac{\sum_{i=1}^{k} p'_i}{\sum_{i=1}^{k} q'_i} \leq \frac{\sum_{i=1}^{k+1} p'_i}{\sum_{i=1}^{k+1} q'_i}$$

and it's sufficient to proving the following inequality:

$$\frac{\sum_{i=1}^{k} p'_i}{\sum_{i=1}^{k} q'_i} \leq \frac{p'_{k+1}}{q'_{k+1}}$$

We then start to prove the inequality by induction.

$k = 1$: Let $a_1 = p'_1, a_2 = p'_2, b_1 = q'_1, b_2 = q'_2$, and by using inequality $\triangle$, the statement is proved.

$k + 1$: By induction:

$$\frac{\sum_{i=1}^{k} p'_i}{\sum_{i=1}^{k} q'_i} \leq \frac{p'_{k+1}}{q'_{k+1}}$$

By leveraging inequality $\triangle$:

$$\frac{\sum_{i=1}^{k+1} p'_i}{\sum_{i=1}^{k+1} q'_i} \leq \frac{p'_{k+1}}{q'_{k+1}}$$

As $\frac{p'_{k+1}}{q'_{k+1}} \leq \frac{p'_{k+2}}{q'_{k+2}}$:

$$\frac{\sum_{i=1}^{k+1} p'_i}{\sum_{i=1}^{k+1} q'_i} \leq \frac{p'_{k+2}}{q'_{k+2}}$$

Thus the lemma is proved. ∎

**Lemma B.3.** *Let $Q \in \mathbb{R}^{d \times d}$ be a rate transition matrix, $A \in \mathbb{R}^{d \times d}$ be a permutation matrix, then $AQA^{-1}$ is a rate transition matrix.*

*Proof.* As every permutation matrix can be expressed as the products of elementary matrices, we denote:

$$A = \prod_{k=N_A}^{1} T_{ij}^{(k)} = T_{ij}^{(N_A)} T_{ij}^{(N_A-1)} \ldots T_{ij}^{(1)}$$

, where $T_{ij}$ is the elementary matrix obtained by swapping row $i$ and row $j$ of the identity matrix, $N_A \in \mathbb{R}$

Therefore:

$$AQA^{-1} = (\prod_{k=N_A}^{1} T_{ij}^{(k)})Q(\prod_{k=1}^{N_A} T_{ij}^{(k)})$$

For a single pair of transformation, i.e. $T_{ij}^{(k)} Q T_{ij}^{(k)}$, the row sums remain unchanged, and the diagonal elements is still the diagonal elements after transformation, thus $AQA^{-1}$ is a rate transition matrix. ∎

### B.2 PROOF OF THE THEOREM

**Theorem B.4** (Accessibility)**.** *For two arbitrary discrete distributions $p, q \in \mathbb{R}^d$, there exists a rate transition matrix $Q \in \mathbb{R}^{d \times d}$ such that:*

$$p = qe^Q$$

*Proof.* By Lemma 4.4, there exists permutation matrix $A$ which satisfies inequality 15, and we denote:

$$p' := pA$$
$$q' := qA$$

Suppose:

$$Q := AQ'A^{-1}$$

, where $Q' = \begin{bmatrix} -\sum_{i=1}^{n-1} a_i & a_1 & a_2 & \ldots & a_{n-2} & a_{n-1} \\ 0 & -\sum_{i=2}^{n-1} a_i & a_2 & \ldots & a_{n-2} & a_{n-1} \\ \ldots & \ldots & \ldots & \ldots & \ldots & \ldots \\ 0 & 0 & 0 & \ldots & -a_{n-1} & a_{n-1} \\ 0 & 0 & 0 & 0 & 0 & 0 \end{bmatrix} = U\Lambda U^{-1}$, $U$ is all one upper

triangle matrix, and $\Lambda = diag(\{-\sum_{i=1}^{n-1} a_i, -\sum_{i=2}^{n-1} a_i, \ldots, -a_{n-1}, 0\})$

Denote:

$$p'' := p'U = [p_1', p_1' + p_2', \ldots, \sum_{i=1}^{n-1} p_i', 1]$$

$$q'' := q'U = [q_1', q_1' + q_2', \ldots, \sum_{i=1}^{n-1} q_i', 1]$$

Thus the solution of $p = qe^Q$ can be obtained by solving:

$$p'' = q''e^\Lambda$$

, where $e^{\Lambda} = diag(\{e^{-\sum_{i=1}^{n-1} a_i}, e^{-\sum_{i=2}^{n-1} a_i}, \ldots, e^{-a_{n-1}}, 1\})$ Solving the equation:

$$a_k = \ln \frac{\sum_{i=1}^{k+1} p'_i}{\sum_{i=1}^{k+1} q'_i} - \ln \frac{\sum_{i=1}^{k} p'_i}{\sum_{i=1}^{k} q'_i}$$

and specifically,

$$a_{n-1} = -\ln \frac{\sum_{i=1}^{n-1} p'_i}{\sum_{i=1}^{n-1} q'_i}$$

By the inequality 15 which $p', q'$ satisfies and the monotonicity of the $\ln()$ function, $a_k \geq 0, \forall k$, and thus $Q'$ is a rate transition matrix

Transfering the solution of $p'' = q'' e^{\Lambda}$ back, we obtain:

$$Q = AU\Lambda U^{-1}A^{-1} = AQ'A^{-1}$$

and by Lemma 4.5, Q is a rate transition matrix. ∎

## C   PROOF OF SUPERVISION OF *Score*-LEARNING

**Proposition C.1** (Supervision of *Score*-learning). *Suppose $\boldsymbol{Q}^{(t)}$'s elements are non-zeros, the training objective is depicted as in Equation (8), then the optimality of the score model $s_{\theta*}(x_t, t)_b$ satisfies:*

$$s_{\theta*}(x_t, t)_y = \mathbb{E}_{x_0 \sim \mu_{0|t}(\cdot|x_t)}\left[\frac{p_{t|0}(y|x_0)}{p_{t|0}(x_t|x_0)}\right] = \frac{\sum_{x_0} \mu(x_0)p_{t|0}(y|x_0)}{\sum_{x_0} \mu(x_0)p_{t|0}(x_t|x_0)}$$

*Proof.*

$$J_{score} = \int_0^T \mathbb{E}_{x_0 \sim \mu, x_t \sim p_{t|0}(x_t|x_0)}\Big[\sum_{y \neq x_t} \boldsymbol{Q}^{(t)}_{y,x_t}\Big(s_{\theta}(x_t, t)_y - \frac{p_{t|0}(y|x_0)}{p_{t|0}(x_t|x_0)}$$

$$+ \frac{p_{t|0}(y|x_0)}{p_{t|0}(x_t|x_0)}\Big(\log s_{\theta}(x_t, t)_y - \log(\frac{p_{t|0}(y|x_0)}{p_{t|0}(x_t|x_0)}))\Big)\Big]dt$$

Therefore, with a little abuse of notation, we have

$$\arg\min_{\theta} J_{score} = \arg\min_{\theta} \int_0^T \mathbb{E}_{x_0 \sim \mu, x_t \sim p_{t|0}(x_t|x_0)}\Big[\sum_{b \neq x_t} \boldsymbol{Q}^{(t)}_{y,x_t}\Big(s_{\theta} - \frac{p_{t|0}(y|x_0)}{p_{t|0}(x_t|x_0)}\log s_{\theta}\Big)\Big]dt$$

$$= \arg\min_{\theta} \underbrace{\int_0^T \mathbb{E}_{x_t \sim \mu_t}\Big[\sum_{y \neq x_t} \boldsymbol{Q}^{(t)}_{y,x_t}\Big(s_{\theta} - \mathbb{E}_{x_0 \sim \mu_{0|t}}[\frac{p_{t|0}(y|x_0)}{p_{t|0}(x_t|x_0)}]\log s_{\theta}\Big)\Big]dt}_{\mathcal{L}}$$

$$\frac{\partial \mathcal{L}}{\partial s_{\theta}} = \int_0^T \mathbb{E}_{x_t \sim \mu_t}\Big[\sum_{y \neq x_t} \boldsymbol{Q}^{(t)}_{y,x_t}\Big(1 - \mathbb{E}_{x_0 \sim \mu_{0|t}}[\frac{p_{t|0}(y|x_0)}{p_{t|0}(x_t|x_0)}]\frac{1}{s_{\theta}}\Big)\Big]dt$$

As $\boldsymbol{Q}^{(t)}$'s elements are non zeros, therefore

$$\boldsymbol{Q}^{(t)}_{y,x_t} > 0, \forall y \neq x_t$$

$$\frac{\partial \mathcal{L}}{\partial s_\theta} = 0 \iff 1 - \mathbb{E}_{x_0 \sim \mu_{0|t}}\left[\frac{p_{t|0}(y|x_0)}{p_{t|0}(x_t|x_0)}\right]\frac{1}{s_\theta} = 0$$

Therefore, the optimality of $s_\theta$ satisfies:

$$s_{\theta^*}(x_t, t)_y = \mathbb{E}_{x_0 \sim \mu_{0|t}(\cdot|x_t)}\left[\frac{p_{t|0}(y|x_0)}{p_{t|0}(x_t|x_0)}\right]$$

Furthermore, as $\mu_{0|t}(x_0|x_t) = \frac{\mu(x_0)p_{t|0}(x_t|x_0)}{\sum_{x_0}\mu(x_0)p_{t|0}(x_t|x_0)}$, we have

$$s_{\theta^*}(x_t, t)_y = \frac{\sum_{x_0}\mu(x_0)p_{t|0}(y|x_0)}{\sum_{x_0}\mu(x_0)p_{t|0}(x_t|x_0)}$$

∎

# D  PROOF OF CONVERGENCE

## D.1  PROOF OF LEMMAS

**Lemma D.1.** *For a random variable $X_0 \in \mathbb{R}^n$ with arbitrary two distributions $p_0, p_0'$, the transition kernel is $p_{t|0}(x_t|x_0)$. We denote*

$$p_t(x_t) := \sum_{x_0} p_0(x_0)p_{t|0}(x_t|x_0)$$

$$p_t'(x_t) := \sum_{x_0} p_0'(x_0)p_{t|0}(x_t|x_0)$$

*Then we have:*

$$D_{KL}(p_t||p_t') \leq D_{KL}(p_0||p_0')$$

*Proof.*

$$
\begin{aligned}
D_{KL}(p_{0,t}(\cdot,\cdot)||p_{0,t}'(\cdot,\cdot)) &= \sum_{x_0,x_t} p_{0,t}(x_0,x_t) \log \frac{p_{0,t}(x_0,x_t)}{p_{0,t}'(x_0,x_t)} \\
&= \sum_{x_0,x_t} p_{0,t}(x_0,x_t) \log \frac{p_{t|0}(x_t|x_0)p_0(x_0)}{p_{t|0}(x_t|x_0)p_0'(x_0)} \\
&= D_{KL}(p_0||p_0')
\end{aligned}
$$

Using the chain rule for KL divergence:

$$D_{KL}(p_t||p_t') = D_{KL}(p_{0,t}(x_0,x_t)||p_{0,t}'(x_0,x_t)) - \mathbb{E}_{p_t}[D_{KL}(p_{0|t}(x_0|x_t)||p_{0|t}'(x_0|x_t)]$$

As KL divergence is greater than zero, we have:

$$D_{KL}(p_t||p_t') \leq D_{KL}(p_{0,t}(x_0,x_t)||p_{0,t}'(x_0,x_t)) = D_{KL}(p_0||p_0')$$

∎

## D.2  PROOF OF THE THEOREM

**Theorem D.2** (Convergence of the algorithm)**.** *If we assume optimality is achieved in every epoch of the forward process and the reverse process, and we denote the $k$-th epoch estimation of $\mu$ as $p_0$, then $\lim_{k\to\infty} D_{KL}(\mu||p_0^{(k)})$ converges.*

*Proof.* According to the assumption that each subprocess reaches its optimum,

$$\mu = \mu p_{T|0}^{(k)} p_{0|T}^{(k);\leftarrow}$$

$$p_0^{(k+1)} = p_0^{(k)} p_{T|0}^{(k)} p_{0|T}^{(k);\leftarrow}$$

Therefore, by using Lemma D.1 twice:

$$D_{KL}(\mu||p_0^{(k)}) \geq D_{KL}(\mu p_{T|0}^{(k)}||p_0^{(k)} p_{T|0}^{(k)}) \geq D_{KL}(\mu p_{T|0}^{(k)} p_{0|T}^{(k);\leftarrow}||p_0^{(k)} p_{T|0}^{(k)} p_{0|T}^{(k);\leftarrow})$$

Therefore,

$$D_{KL}(\mu||p_0^{(k)}) \geq D_{KL}(\mu||p_0^{(k+1)})$$

As KL divergence is greater than zero, then

$$\lim_{k \to \infty} D_{KL}(\mu||p_0^{(k)})$$

converges. ∎

## E    DERIVATION OF MATRIX EXPONENTIAL CALCULATION

**Proposition E.1.** *For a matrix $Q \in \mathbb{R}^{n \times n}$ and a non-degenerate matrix $D \in \mathbb{R}^{n \times n}$, we have:*

$$\exp\{DQD^{-1}\} = D\exp\{Q\}D^{-1}$$

*Proof.* According to the definition of matrix exponential,

$$\exp\{DQD^{-1}\} = I + \sum_{i=1}^{\infty}(DQD^{-1})^i$$

As $(DQD^{-1})^i = DQ^i D^{-1}$,

$$\exp\{DQD^{-1}\} = I + \sum_{i=1}^{\infty} DQ^i D^{-1} = D(I + \sum_{i=1}^{\infty} Q^i)D^{-1} = D\exp\{Q\}D^{-1}$$

∎

## F    DERIVATION OF KL TERM CALCULATION PROPOSITION

The full bound (Meng et al., 2023; Campbell et al., 2022) is as follows:

$$\mathbb{E}_{x_0 \sim \mu}[-\log p_{0;\theta}(x_0)] \leq J_{score} + \mathbb{E}_{x_0 \sim \mu}[D_{KL}\big(p_{T|0;\alpha}(x_T|x_0)||\phi\big)]$$

, where

$$J_{score} \triangleq \int_0^T \mathbb{E}_{x_0 \sim \mu, x_t \sim p_{t|0}(x_t|x_0)}\Big[\sum_{b \neq x_t} Q_{b,x_t}^{(t)}\bigg(s_\theta(x_t,t)_b - \frac{p_{t|0}(b|x_0)}{p_{t|0}(x_t|x_0)}$$

$$+ \frac{p_{t|0}(b|x_0)}{p_{t|0}(x_t|x_0)}\big(\log s_\theta(x_t,t)_b - \log(\frac{p_{t|0}(b|x_0)}{p_{t|0}(x_t|x_0)}))\bigg)\Big]dt$$

However, unlike previous works, the second term, which is the $KL$ term should be considered, and it seems impossible to compute. Fortunately, certain characteristics of the *Matrix*-learning process can be used to justify a computable form for the second term. Suppose the text sequence holds $d$ dimensions, $i.e. x \in \mathbb{R}^d$, then the characteristics can be described as follows:

- Independent Evolution:

$$p_{T|0;\alpha}(x_T|x_0) = \prod_{i=1}^{d} p_{T|0;\alpha}(x_T^{(i)}|x_0^{(i)})$$

- Independent Terminal:

$$\phi(x_T) = \prod_{i=1}^{d} p_T(x_T^{(i)})$$

As a result, we provide a computable form for the KL term.

**Proposition F.1.** $D_{KL}\big(p_{T|0;\alpha}(x_T|x_0)||p_T(x_T)\big) = \sum\limits_{i=1}^{d} D_{KL}\big(p_{T|0;\alpha}(x_T^{(i)}|x_0^{(i)})||p_T(x_T^{(i)})\big)$

*Proof.* By independent evaluation and independent terminal, we have

$$D_{KL}\big(p_{T|0;\alpha}(x_T|x_0)||p_T(x_T)\big) = \sum_{x_T} p_{T|0;\alpha}(x_T|x_0) \log \frac{p_{T|0;\alpha}(x_T|x_0)}{\phi}$$

$$= \sum_{x_T^{(1)},x_T^{(2)},...,x_T^{(d)}} p_{T|0;\alpha}(x_T|x_0) \sum_{i=1}^{d} \log \frac{p_{T|0;\alpha}(x_T^{(i)}|x_0^{(i)})}{p_T(x_T^{(i)})}$$

$$= \sum_{i=1}^{d} \sum_{x_T^{(1)},x_T^{(2)},...,x_T^{(d)}} p_{T|0;\alpha}(x_T|x_0) \log \frac{p_{T|0;\alpha}(x_T^{(i)}|x_0^{(i)})}{p_T(x_T^{(i)})}$$

$$= \sum_{i=1}^{d} \sum_{x_T^{(i)}} p_{T|0;\alpha}(x_T^{(i)}|x_0^{(i)}) \log \frac{p_{T|0;\alpha}(x_T^{(i)}|x_0^{(i)})}{p_T(x_T^{(i)})}$$

$$= \sum_{i=1}^{d} D_{KL}\big(p_{T|0;\alpha}(x_T^{(i)}|x_0^{(i)})||p_T(x_T^{(i)})\big)$$

∎

## G ADDITIONAL EXPERIMENTAL DETAILS

### G.1 MODEL DETAILS

In terms of text modeling, for *Matrix*-learning, the $Q_\alpha$ matrix is initialized as follows:

$$a_i = 0, \forall i = 1, 2, 3, \ldots, n-2$$

$$a_{n-1} = 1$$

The model is kept the same as SEDD Lou et al. (2024).

As for image modeling, for *Matrix*-learing, the $Q_\alpha$ matrix is initialized as follows:

$$a_i = 1e-5, \forall i = 1, 2, 3, \ldots, n-2$$

The model is kept the same as SEDD Lou et al. (2024).

### G.2 TRAINING DETAILS

The model is trained with a batch size of $512$ and trained with a learning rate of $3 \times 10^{-4}$ (Adam optimizer) on $8$ 4090 24GB GPUs. Both the *Matrix*-learning as well as the *Score*-learning are trained with the AdamW Loshchilov & Hutter (2019). Training start with a weight decay factor 0.01, which then turn to 0 in the 7,900,000 step for text8.

## H DISCUSSION AND FUTURE WORKS

In this work, the **DMB** framework primarily relies on the evidence lower bound (ELBO) for both training and evaluation. However, given that Theorem 4.7 is not dependent on the specific form of the loss function, it is theoretically possible to derive other bounds for training. This flexibility opens new avenues for optimizing **DMB** under different theoretical and practical settings. Furthermore, we haven't provided a theorem focusing on optimality, which may be done for future work.

# I  THE USE OF LARGE LANGUAGE MODELS (LLMs)

The use of Large Language Models (LLMs) in this work was confined to editorial tasks, namely improving the manuscript's prose and generating figures. All core scientific contributions—including the DMB framework, its theoretical development, experimental design, and analysis of the results—are the exclusive work of the authors.

