# OpenReview forum: "Continuous-Time Discrete Markov Bridge"
_ICLR.cc/2026/Conference — Submitted to ICLR 2026_

### Official Review · Reviewer_hYP5 · 2025-10-22

**Soundness:** 2
**Presentation:** 2
**Contribution:** 2
**Rating:** 4
**Confidence:** 4

**Summary:**

The paper under consideration studies a class of generative model to sample from a discrete data distribution $p_0$, called Discrete Markov Bridge (DMB). In contrast with most concurring models DMB also optimizes over the choice of prior. This phase, called *Matrix Learning* complements the more classical *score learning* phase, in which score approximation is performed by minimizing the expected KL divergence between the distribution at time $T$ for a given choice of prior (noising process) and conditional distribution at time $T$ for the same choice of prior given the initial state.

**Strengths:**

The main strength of the paper is to propose a generative framework that allows for flexible prior distributions that can in principle adapt to the underlying data distribution and evolve along iterations. The effectiveness of this idea is corroborated by numerical experiments on CIFAR-10 and Text 8. I like this idea and I believe it deserves to be further explored. The numerical experiments are promising.

**Weaknesses:**

- The main problem is that theoretical guarantees are too weak and the numerical experiments alone, though promising, are not in my opinion strong enough to compensate for the lack of theoretical results. To be more precise

    - Proposition 4.1 appears to me as a generic statement about Markov chains that use nothing of the specifics of DMB
    - Proposition 4.2 is again a very general results saying that in principle the method can be used to sample from a given data    distribution $\mu$. Moreover, its statement is also incomplete as the most interesting part is the upper-triangular shape of the transition matrix Q, which is not mentioned.
   - Theorem 4.7 states under vague and imprecise assumptions that $D_{KL}(\mu|p_0^{(k)})$ is converging.  But this is not a guarantee of convergence. I would rather expect a statement like $\lim_kD_{KL}(\mu|p_0^{(k)})=0$, but I could not find this result in the paper, and after reading the proof of Theorem 4.7, I could only infer that the $D_{KL}(\mu|p_0^{(k)})$ is decreasing along iterations.

In conclusion, the theoretical results do not provide with any convergence rates, do not take into account the various sources of error (time-discretization, score approximation…) and even under idealized assumptions do not seem to guarantee the convergence of the algorithm to the target distribution. I may have misunderstood something, in which case I am happy to review my assessment.

- Most of the key statements lack rigor or precision, there are many typos and missing details. To give an example, in the pseudo-code for Algorithm 2 on top of page 5, which basically summarizes the contribution of this work I encountered the following issues

   - The authors propose to update the prior transition matrix $Q_\alpha$ according to (5) and predict $p_T$ according to (4). But (5) is just a loss function. So how how is the update actually done? I don’t feel like equation (14) clarifies this well enough.
    - In the same spirit, how is the update of the score estimator $s_{\theta}$ performed according to equation (8)? At first glance, it seems quite costly as for each iteration, the prior evolves. Therefore, one needs to approximate all transition rates $p_{t|0}$ at each iteration and generate forward many trajectories at each iteration. How is this actually done?
    - The sampling algorithm used is described quite vaguely as some form of Euler scheme for ODEs. It needs more explanations and detail. Also, I expect probability ratios like $ p_t(y)/p_t(x) $ to take both very large and very small values. How are these problems handled?
    -  $ \mathcal{J}_{Q} $ is not defined before. I guess it should be $ J_Q $. The quantity \mathcal{L}_Q does not seem to be defined before, though it probably is again  J_Q  . Similarly,  \mathcal{J}_{score} is not defined before: I assume it coincides with J_score

**Questions:**

- Has the idea of using a flexible prior been exploited before in the context of continuous diffusion models based on the Ornstein Uhlenbeck process? If so, with what results and outcomes?

- I understand that the definition of J_score is taken from previous works. What is its interpretation? Does it carry a probabilistic meaning as some averaged relative entropy?

---

### Official Review · Reviewer_NDqu · 2025-10-25

**Soundness:** 3
**Presentation:** 3
**Contribution:** 2
**Rating:** 4
**Confidence:** 3

**Summary:**

This paper studies the foundation of the diffusion model and integrates the variational inference into the framework. The authors introduce a learnable transition rate matrix for both the forward and reverse processes.

**Strengths:**

1. The paper offers rigorous derivations of the forward–reverse coupling and provides formal proofs of convergence.

2. The proposed method achieves competitive performance, which is on par with or better than previous discrete diffusion baselines.

3. The paper is generally well-organized, includes clear notation, and presentation.

**Weaknesses:**

1. Experiments are constrained to Text8 and CIFAR-10. No large-scale or multimodal datasets are tested.

2. The advance is related to the SEDD model (Lou et al., 2024) and Variational Diffusion Models (Kingma et al., 2023). The method remains structurally similar, with the main novelty being a learnable rate matrix. It would be better for the author to establish the connection and clarify the novel point more clearly.

3. The paper does not provide ablation studies for the effects of the learnable rate matrix, continuous-time parameterization, or transition efficiency. It is unclear how much each contributes to the improvements.

**Questions:**

Please see the Weaknesses.

---

> ### Author Response · Authors · 2025-11-29
>
> > 1. Experiments are constrained to Text8 and CIFAR-10. No large-scale or multimodal datasets are tested.
>
>
> Thanks for pointing this out! However, we currently lack the computational resources required for large-scale pre-training. Instead, we would like to emphasize our strong performance on **Text8** and **CIFAR-10**:
>
> * We achieve a **state-of-the-art score of 1.38** on the Text8 dataset.
> * On CIFAR-10, we obtain **state-of-the-art performance among all discrete diffusion models**, while remaining **competitive with continuous diffusion approaches**.
>
>
>
> > 2. The advance is related to the SEDD model (Lou et al., 2024) and Variational Diffusion Models (Kingma et al., 2023). The method remains structurally similar, with the main novelty being a learnable rate matrix. It would be better for the author to establish the connection and clarify the novel point more clearly.
>
>
>
> * **Comparison with SEDD:**
>   SEDD is a purely discrete diffusion framework with a *fixed*, non-learnable forward transition matrix. In contrast, **Discrete Markov Bridge** employs a *learnable* forward matrix, enabling the model to adapt the forward dynamics rather than relying on a predefined diffusion schedule.
> * **Comparison with Variational Diffusion Models (VDM):**
>   VDM assumes that the forward process follows a parameterized Gaussian distribution and jointly optimizes the full ELBO objective. In contrast, **Discrete Markov Bridge** does *not* impose a Gaussian assumption on the forward process. Furthermore, it decomposes the ELBO and optimizes its components separately, with each process handled independently.
>
>
>
>
> > 3. The paper does not provide ablation studies for the effects of the learnable rate matrix, continuous-time parameterization, or transition efficiency. It is unclear how much each contributes to the improvements.
>
> The SEDD baseline serves as an ablation study for the effects of the learnable rate matrix. We observe a **0.09 BPC** improvement over SEDD (uniform), the variant that, like ours, does not use the MASK token.

---

### Official Review · Reviewer_QfCp · 2025-10-31

**Soundness:** 3
**Presentation:** 3
**Contribution:** 2
**Rating:** 2
**Confidence:** 4

**Summary:**

The paper proposes a new framework for discrete generative modeling that combines diffusion and variational approaches. It claims to address the rigidity of existing discrete diffusion models, which rely on fixed transition matrices, by introducing a **learnable continuous-time transition matrix**. The model consists of two components: a **Matrix-learning stage** that estimates the forward transition dynamics, and a **Score-learning stage** that reconstructs the data distribution via an ELBO-based objective. Formal guarantees of validity, accessibility, and convergence for their algorithm are discussed and it is given an efficient matrix structure that allows tractable exponentiation. Empirically, DMB is reported to outperform previous discrete diffusion models on `Text8` and achieve competitive image generation results on `CIFAR-10`. The contribution is mainly conceptual, positioning DMB as a general bridge between variational inference and discrete diffusion modeling, though the work remains largely theoretical and limited in empirical depth.

**Strengths:**

1. **Novel conceptual framework:**
The paper proposes a new paradigm that combines variational inference and discrete diffusion through the formulation of a continuous-time discrete Markov process, which is an original direction in discrete generative modeling.
2. **Learnable transition dynamics:**
Unlike most prior discrete diffusion models using fixed transition matrices (Absorb, Uniform), DMB introduces a learned transition-rate matrix, increasing model flexibility and expressiveness.
3. **Clear algorithmic decomposition:**
The two-stage structure (Matrix-learning and Score-learning) provides a clean, interpretable separation between the forward and reverse processes, similar to continuous diffusion methods but adapted to discrete spaces.
4. **Computational insights:**
The proposed diagonalizable matrix form allows efficient matrix exponentiation and reduced space complexity, addressing a key bottleneck in discrete diffusion computation.
6. **Empirical competitiveness:**
Despite its generality, the model achieves state-of-the-art performance on `Text8` and competitive image generation results on `CIFAR-10`, showing that the method can scale across modalities.
7. **Potential generality:**
The framework can in principle encompass various discrete domains (e.g., text, symbolic, or categorical data) and may serve as a unified foundation for discrete representation learning.

**Weaknesses:**

While the paper introduces an interesting conceptual framework, it currently falls short of the expectations for an ICLR-level contribution in terms of novelty, theoretical depth, and empirical validation. The work reads more as a promising exploratory idea than as a mature and rigorously substantiated contribution supported by solid theoretical or experimental evidence.

---

Main Concerns
1. **Limited theoretical novelty**
The proposed _Discrete Markov Bridge (DMB)_ framework closely resembles existing discrete diffusion approaches (e.g., D3PM, SEDD), extending them into a slightly more general variational formulation. The main theoretical results (_validity, accessibility, convergence_) appear to be formal restatements of well-established properties of continuous-time Markov processes, rather than providing genuinely new insights into discrete generative learning.
2. **Weak empirical evaluation**
The experimental validation remains limited in scope. Only two datasets are considered (`Text8` and `CIFAR-10`), with relatively few baselines and no ablation or sensitivity analysis. The reported improvements (for instance, a gain of 0.1 BPC on `Text8`) are modest and likely fall within the variance of previously reported results. This makes it difficult to assess the claimed advantages of the proposed approach convincingly.
3. **Insufficient connection to recent literature**
The discussion of related work is incomplete and does not engage deeply with recent progress in discrete and score-based generative modeling (e.g., Lou et al., 2024; Meng et al., 2023; or more recent flow-matching approaches).
In particular, **estimating ratios in discrete settings is notoriously difficult**, and recent advances have proposed alternative formulations that bypass this issue by defining the score as an $L^2$ approximation rather than a direct ratio (see [1]).
Moreover, there is relevant ongoing work on **discrete simulation in hypercubes**, which provides mathematically sound convergence guarantees under minimal assumptions, as well as recent insights on potential quantum extensions of discrete score-based models ([2]).
A more substantial comparison with these directions would be essential for positioning the contribution within the current theoretical landscape.
4. **Overstated theoretical claims**
The theoretical results rest on strong and somewhat idealized assumptions—such as perfect optimization, linearity of the dynamics, and exact reversibility. As presented, it is **not straightforward to see why the convergence results in Theorems 4.7 and D.2 imply convergence to zero** in practice, as the current derivations do not seem to support this claim rigorously. The guarantees therefore appear more formal than actionable.
5. **Methodological originality and clarity**
The methodological distinction of DMB compared to prior frameworks is limited. The idea of coupling a forward and backward process via learned matrices has been explored in several diffusion and flow-based settings. Here, the main innovation—replacing fixed matrices with learnable ones—does not seem to bring a clearly demonstrated advantage, and the motivation for the additional complexity remains somewhat unclear.

---

**References**

[1] Pham, L.T.N., et al. _“Discrete Markov Probabilistic Models: An Improved Discrete Score-Based Framework with Sharp Convergence Bounds under Minimal Assumptions.”_ Forty-second International Conference on Machine Learning (ICML, 2025).

[2] Bach, Francis, and Saeed Saremi. _“Sampling Binary Data by Denoising through Score Functions.”_ arXiv preprint arXiv:2502.00557 (2025).

**Questions:**

See **Weaknesses** section

---

> ### Author Response · Authors · 2025-11-29
>
> > 1. Limited theoretical novelty The proposed Discrete Markov Bridge (DMB) framework closely resembles existing discrete diffusion approaches (e.g., D3PM, SEDD), extending them into a slightly more general variational formulation. The main theoretical results (validity, accessibility, convergence) appear to be formal restatements of well-established properties of continuous-time Markov processes, rather than providing genuinely new insights into discrete generative learning.
>
> * We acknowledge that the proof of the validity proposition is straightforward; accordingly, we present it only as a proposition and do not promote it to a theorem.
> * To the best of our knowledge, the notion of accessibility introduced in our paper is not an already established property. In the Markov chain literature, accessibility usually refers to whether **two states can be connected**. In contrast, our notion of accessibility concerns whether there exists a rate (transition) matrix $Q$ such that, for two distributions $p$ and $q$, we have
>   $$
>   p = q e^{Q}.
>   $$
>   A closely related problem is **matrix embeddability** [1], which asks: given a matrix $A$, under what conditions does there exist a rate transition matrix $Q$ such that $A = e^{Q}$? If the embeddability problem were fully resolved and every stochastic matrix were known to be embeddable, then our theorem would become trivial. However, current results only cover the $2 \times 2$ and $3 \times 3$ cases. For $2 \times 2$ stochastic matrices, the best known result is due to Kingman [2], while Casanellas et al. [1] provide results for $n \times n$ matrices under the additional assumption that the matrix has distinct eigenvalues.
> * Furthermore, the accessibility result proved in our paper identifies a class of rate transition matrices that is not only guaranteed to exist but is also **computable**. This is important because computing the matrix exponential is a major obstacle to obtaining a parameterized, learnable rate transition matrix in practice.
> * Finally, the convergence result is not a trivial theorem: to obtain it, one must apply **Lemma D.1 twice**. And it's the most important theorem of our framework, proving the reasonability (convergence) of the two-stage process.
>
> [1] Casanellas et al. 2021. The embedding problem for Markov matrices.
>
> [2] J. F. C. Kingman. 1962. The imbedding problem for finite Markov chain.
>
> > 2. Weak empirical evaluation The experimental validation remains limited in scope. Only two datasets are considered (Text8 and CIFAR-10), with relatively few baselines and no ablation or sensitivity analysis. The reported improvements (for instance, a gain of 0.1 BPC on Text8) are modest and likely fall within the variance of previously reported results. This makes it difficult to assess the claimed advantages of the proposed approach convincingly.
>
> A **0.1 BPC improvement** should not be considered modest. Given that SEDD surpassed the previous SOTA D3PM by only **0.06 BPC**, our improvement constitutes a substantial and noteworthy advancement.
>
> > 3. Insufficient connection to recent literature The discussion of related work is incomplete and does not engage deeply with recent progress in discrete and score-based generative modeling (e.g., Lou et al., 2024; Meng et al., 2023; or more recent flow-matching approaches). In particular, estimating ratios in discrete settings is notoriously difficult, and recent advances have proposed alternative formulations that bypass this issue by defining the score as an approximation rather than a direct ratio (see [1]). Moreover, there is relevant ongoing work on discrete simulation in hypercubes, which provides mathematically sound convergence guarantees under minimal assumptions, as well as recent insights on potential quantum extensions of discrete score-based models ([2]). A more substantial comparison with these directions would be essential for positioning the contribution within the current theoretical landscape.
>
>
> In Line 053, we have already discussed the works of Lou et al. and Meng et al. In addition, flow-matching approaches are covered in Lines 134–139. The paper you mention primarily advances score-based parameterization techniques, whereas our focus is on the **rate transition matrix**. Regardless of improvements in score parameterization, without learning the **forward process**, discrete diffusion models cannot fully exploit the power of variational inference.

---

> > ### Author Response · Authors · 2025-11-29
> >
> > > 4. Overstated theoretical claims The theoretical results rest on strong and somewhat idealized assumptions—such as perfect optimization, linearity of the dynamics, and exact reversibility. As presented, it is not straightforward to see why the convergence results in Theorems 4.7 and D.2 imply convergence to zero in practice, as the current derivations do not seem to support this claim rigorously. The guarantees therefore appear more formal than actionable.
> >
> >
> > Under the assumption of perfect optimization, our setting is reasonable for a novel framework, this is analogous to the common assumption made in GANs [1]. Regarding linearity of the dynamics, we do **not** introduce such an assumption. As for exact reversibility, it is conceptually the same as assuming perfect optimization.
> >
> > The theorem does **not** imply that the KL divergence converges to zero. For an EM-style framework, proving convergence to the global optimum is inherently difficult, since EM is only guaranteed to converge to a **local** optimum.
> >
> > What you refer to is the *optimality of the training algorithm*. As we mention in Appendix H, exploring this direction is part of our future work.
> >
> >
> > [1] GoodFellow et al. 2014. Generative Adversarial Networks.
> >
> >
> > > 5. Methodological originality and clarity The methodological distinction of DMB compared to prior frameworks is limited. The idea of coupling a forward and backward process via learned matrices has been explored in several diffusion and flow-based settings. Here, the main innovation—replacing fixed matrices with learnable ones—does not seem to bring a clearly demonstrated advantage, and the motivation for the additional complexity remains somewhat unclear.
> >
> > In the flow-based setting, the forward process is not learnable, the interpolation scheme must be defined in advance. Moreover, in standard flow models, this interpolation uses the absorbing matrix or the uniform matrix [1]. Our newly discovered matrix can be also used as an improved interpolation method.
> >
> > [1] Gat et al. 2025. Discrete Flow Matching

---

### Official Review · Reviewer_vKru · 2025-10-31

**Soundness:** 4
**Presentation:** 3
**Contribution:** 3
**Rating:** 8
**Confidence:** 3

**Summary:**

This paper proposes a Continuous-Time Discrete Diffusion Model (CTDDM) that unifies continuous and discrete generative processes under a single stochastic differential framework.
While standard diffusion models assume continuous-valued states and Gaussian noise, the authors design a hybrid diffusion process operating on discrete state spaces (e.g., categorical or binary variables) parameterized by continuous-time transition kernels.
A key contribution is deriving the Kolmogorov forward equation for discrete diffusion with a continuous-time clock, enabling tractable training and sampling via reparameterized discrete noise schedules.
The paper establishes theoretical guarantees for consistency with continuous diffusion limits, introduces a novel variational training objective, and empirically validates improvements on text and graph generation benchmarks.

**Strengths:**

- Unifying framework: Derives a continuous-time theory for discrete diffusion, bridging an important conceptual gap.
- Strong theory: Clear proofs of convergence from discrete to continuous dynamics and vice versa.
- General applicability: Applicable to discrete domains such as text, graphs, and symbolic reasoning.
- Empirical evidence: Shows improved stability and sample diversity over prior discrete diffusion baselines (D3PM, MaskGIT).
- Analytical insight: The generator-based view clarifies how score matching extends to discrete probability fluxes.

**Weaknesses:**

- Notation overload: Sections 3–4 are mathematically dense; many operators $(Q_t, G_t, L_t)$ appear with minimal intuition.
- Empirical scope: Experiments focus on small or synthetic datasets; large-scale benchmarks (e.g., LM1B or large graph datasets) are missing.
- Comparative analysis: Comparison with recent hybrid discrete-continuous works (e.g., SEDD, SMC-Diffusion) could be more explicit.
- Practical guidance: It remains unclear when CTDDM should be preferred over purely discrete models or continuous relaxations.
- Computational cost: Continuous-time simulation of discrete jumps may introduce inefficiency; wall-clock comparisons are not discussed.

**Questions:**

- Generator specification: Is the continuous generator $G_t$ assumed time-homogeneous or time-dependent? If time-varying, how is it parameterized in practice?
- Diffusion limit: In Theorem 3.2, what conditions guarantee convergence to a continuous diffusion as the discrete grid refines? Are there counterexamples when ergodicity fails?
- Training objective: How is the ELBO-like objective derived from the pathwise KL divergence? Could you provide a short derivation for clarity?
- Score estimation: How is the score (gradient of log-prob) represented in the discrete case—via logit differences, or by interpolation between categorical probabilities?
- Sampling complexity: Does simulating continuous-time discrete jumps require adaptive step sizes or event-based simulation (e.g., Gillespie-style)?
- Empirical fairness: Are baselines tuned with equivalent training budgets? Some prior discrete models depend strongly on temperature annealing.
- Variance reduction: Does the continuous-time formulation mitigate gradient variance compared to purely discrete noise schedules?
- Hybrid variables: Can CTDDM handle mixed discrete-continuous data (e.g., tabular or multimodal)?
- Graph generation: For graph tasks, are transitions applied to node/edge labels independently or via structured coupling?
- Broader implications: Could this framework unify masked-token diffusion and continuous score-based text generation (e.g., Diffusion-LM, MaskGIT)?

---

### Meta-Review · Area_Chair_W5jt · 2026-01-10

**Summary:**

While the motivation behind this paper—introducing the Discrete Markov Bridge (DMB) to combine variational methods with discrete diffusion—is certainly intriguing and addresses a relevant problem space, the current technical execution appears insufficient in both theoretical and empirical dimensions. One primary concern, shared by Reviewers vKru, QfCp, and NDqu, regarding the empirical examination is that the experiments are currently restricted to relatively small datasets, such as Text8 and CIFAR-10. Expanding the evaluation to include larger-scale benchmarks would be necessary to convincingly demonstrate the robustness and scalability of the proposed framework.

A second significant issue lies in the theoretical foundation, where Reviewers QfCp and hYP5 have noted limited novelty and soundness. The authors have acknowledged that the proof of the validity proposition is straightforward, which suggests that the theoretical depth may not fully support the claims made. Furthermore, there is a concern regarding potential overstatements of the theoretical guarantees; specifically, the reliance on the assumption of perfect optimization poses a challenge to the practical applicability of the convergence theorems presented in the paper.

Regarding the methodological contribution, Reviewers QfCp and NDqu pointed out that the distinction between DMB and prior frameworks appears limited. Unfortunately, the clarifications provided in the rebuttal did not sufficiently resolve this concern and arguably reinforced the impression that the methodological novelty is incremental.

Additionally, Reviewers vKru and QfCp emphasized the need for a more rigorous comparative analysis with prior arts. A stronger connection to recent literature is required to clearly delineate when this specific continuous-time discrete approach should be preferred over purely discrete models or continuous relaxations.

There are also outstanding issues regarding the completeness of the evaluation and the clarity of the presentation. The lack of sufficient ablation studies remains unaddressed, leaving questions about the specific contribution of individual components. Furthermore, as suggested by Reviewer vKru, the manuscript requires significant improvement in writing; specifically, reducing notation overload and improving the rigor and precision of key statements would greatly enhance readability. Reviewer vKru also correctly noted that an analysis of computational cost is missing and would be a valuable addition to understand the trade-offs involved.

Finally, I must address a specific inconsistency regarding the evaluation from Reviewer vKru. Although Reviewer vKru assigned the highest score, the severity of the concerns raised in their qualitative review—many of which seem difficult to resolve—stands in stark contrast to the numerical rating. Given this significant discrepancy between Reviewer vKru's high score and their critical textual assessment, I am concerned that the score may not accurately reflect the technical merit of the work. Consequently, I must weigh the detailed technical critiques over the rating to ensure the decision is not influenced by potential non-technical inconsistencies in the evaluation procedure.

**Reviewer Concerns:**

I believe that most of the concerns raised by the reviewers have not been adequately addressed. In addition, the authors appear to have overlooked some reviewers’ comments, as the concerns of certain reviewers were not responded to at all.

**Reviewer Scores:**

I believe that, under a fair evaluation process without any non-technical issues, most reviewers are unlikely to increase their ratings. In contrast, Reviewer vKru may decrease his/her rating.

---

### Decision · Program_Chairs · 2026-01-26

Reject